# Analysis of the Accuracy of Ten Algorithms for Orientation Estimation Using Inertial and Magnetic Sensing under Optimal Conditions: One Size Does Not Fit All

**DOI:** 10.3390/s21072543

**Published:** 2021-04-05

**Authors:** Marco Caruso, Angelo Maria Sabatini, Daniel Laidig, Thomas Seel, Marco Knaflitz, Ugo Della Croce, Andrea Cereatti

**Affiliations:** 1Polito^BIO^Med Lab—Biomedical Engineering Lab and Department of Electronics and Telecommunications, Politecnico di Torino, 10129 Torino, Italy; marco.knaflitz@polito.it (M.K.); andrea.cereatti@polito.it (A.C.); 2Department of Excellence in Robotics & AI, The BioRobotics Institute, Scuola Superiore Sant’Anna, 56127 Pisa, Italy; angelo.sabatini@santannapisa.it; 3Control Systems Group, Technische Universität Berlin, 10623 Berlin, Germany; laidig@control.tu-berlin.de (D.L.); seel@control.tu-berlin.de (T.S.); 4Department of Biomedical Sciences, University of Sassari, 07100 Sassari, Italy; dellacro@uniss.it

**Keywords:** MIMU, orientation estimation, filter parameters, filter comparison, wearable sensors, sensor fusion, human motion, Kalman filters, complementary filters, optimal parameters

## Abstract

The orientation of a magneto and inertial measurement unit (MIMU) is estimated by means of sensor fusion algorithms (SFAs) thus enabling human motion tracking. However, despite several SFAs implementations proposed over the last decades, there is still a lack of consensus about the best performing SFAs and their accuracy. As suggested by recent literature, the filter parameters play a central role in determining the orientation errors. The aim of this work is to analyze the accuracy of ten SFAs while running under the best possible conditions (i.e., their parameter values are set using the orientation reference) in nine experimental scenarios including three rotation rates and three commercial products. The main finding is that parameter values must be specific for each SFA according to the experimental scenario to avoid errors comparable to those obtained when the default parameter values are used. Overall, when optimally tuned, no statistically significant differences are observed among the different SFAs in all tested experimental scenarios and the absolute errors are included between 3.8 deg and 7.1 deg. Increasing the rotation rate generally leads to a significant performance worsening. Errors are also influenced by the MIMU commercial model. SFA MATLAB implementations have been made available online.

## 1. Introduction

The accurate estimation of the orientation of a rigid body from the recordings of miniaturized low-cost magneto-inertial measurement units (MIMUs) is still an open challenge for the human movement analysis community. Errors affecting the orientation estimates have a direct negative impact on the quality of estimated quantities in both angular and linear kinematics, therefore limiting the full exploitation of inertial sensing in monitoring daily-life physical activities, as well as in clinical and sports applications [1].

In its full configuration, a MIMU embeds a triaxial accelerometer which measures the specific force (i.e., the vector difference between the coordinate and the gravity accelerations), a triaxial gyroscope which measures the angular rate, and a triaxial magnetometer which senses the local magnetic field (i.e., the vector sum between the Earth’s magnetic field and the external magnetic fields created by ferromagnetic disturbances). The sensor fusion approach aims at estimating the absolute orientation of the MIMU with respect to a global coordinate system (GCS), usually defined to have a vertical axis aligned with the gravity direction and one horizontal axis direction aligned with the Earth’s magnetic north, by exploiting the complementary characteristics of the signals recorded by the MIMU. The first step consists in integrating the kinematics equation which links the angular rate with the orientation change over time to obtain a first approximation of the orientation estimate. The initial conditions for the integration can be obtained by an absolute orientation estimate by using only the accelerometer and the magnetometer measurements in absence of motion [2]. However, the orientation estimated this way is prone to drift due to the integration of the slow-varying bias affecting the gyroscope measurements [3]. To cope with this problem the accelerometer and the magnetometer measurements are employed to correct the drift in both roll and pitch angles (also jointly known as inclination) and yaw angle (also known as declination or heading). Nonetheless, such correction shows some limitations. In fact, the inclination estimated by the accelerometer is highly reliable only during static conditions (i.e., when the coordinate acceleration is negligible and only the gravity direction is sensed). Moreover, the heading resulting from the magnetometer measurements needs to be dealt with care due to the presence of external magnetic fields. Researchers from different fields such as navigation and biomechanics have proposed several sensor fusion implementations over the years, including machine and deep learning approaches, to provide accurate orientation estimates using MIMUs [2,3,4,5,6,7,8,9,10,11,12,13,14,15,16,17,18,19,20,21,22,23]. The large majority of the published sensor fusion algorithms (SFAs) can be grouped in two main classes: Kalman filters (KF) [24] and complementary filters (CF). In the last decades, several formulations of both classes have been proposed including different mathematical orientation representations (e.g., quaternion, rotation matrix, Euler angles etc.), different Kalman filter formulations (direct or indirect, linear, extended, unscented, etc.), and different strategies to fuse the signal information (algebraic or optimization) [15]. Despite the large number of studies aimed at comparatively evaluating different sensor fusion algorithms [2,7,8,9,14,15,16,17,18,22,25,26,27,28,29] and type of sensors (SFA inter-consistency) [30], contradictory results have been observed and the literature is still inconclusive about the expected level of accuracy associated to the MIMU orientation estimation.

Based on the existing body of literature, it is difficult to draw conclusions regarding the “best” algorithm and filtering approach (e.g., CF or KF). Furthermore, errors appear to be highly variable depending on the experimental scenario, commercial device, and algorithm, thus making the generalization of the results impossible [18]. A way of looking at the body of literature is to group studies proposing novel algorithms (“original algorithm studies”) ([2,7,8,9,14,15,16,17,18,22]) and studies focusing on the comparison of existing algorithms/software packages without proposing new ones (“comparative studies”) ([25,26,27,28]). A summary of the results in “comparative studies” and “original algorithm studies” are reported in the Appendix A (Table A1 and Table A2). As a general observation, the magnitude of the errors reported in “comparative studies” are usually higher than those reported in the “original algorithm studies”, up to one order of magnitude. For example, the errors for Madgwick’s filter reported in the original study [9] amounted to about 1 deg, while in [25,28], which involved more challenging experimental conditions, errors were greater than 13 deg. Furthermore, the filter proposed by Guo et al., in [17] was compared with the KF by Valenti et al. [13] for which the errors were twice as large than those reported in the original article. These differences may be due to a number of reasons. First, in the “comparative studies” the experimental conditions under which SFAs are tested are often different from those employed in the original paper in terms of hardware, sensor noise, rotation rates, accelerations magnitude, ferromagnetic disturbances, type of motion, etc. In addition, when a new SFA is presented and its performance evaluated, the proponents often know how to optimally tune the SFA for the specific operating conditions based either on ground-truth knowledge or following a trial-and-error approach [31]. However, when non-experts apply SFAs to different experimental datasets, or in specific human movement applications, the SFA optimization can be difficult resulting in a performance deterioration. It has often been observed that feeding the proper parameter values to any SFA is crucial [22,27,32]. Several intrinsic and extrinsic factors affect the choice of the parameter values, among them the most influencing are amplitude of motion, sensors noise specification, time required by the algorithm to reach convergence, and the amount of ferromagnetic disturbances [31,33].

Based on the considerations above, it is evident that carrying out any “fair”, meaningful and generalizable comparative evaluation among SFAs requires the SFAs parameters to be properly tuned and used under identical experimental conditions (i.e., using the same dataset).

The primary aim of this work is to perform a thorough comparative evaluation of the accuracy of ten of the most popular SFAs proposed in the literature by considering experimental data recorded by three commercial products (Xsens-MTx (Xsens, Enschede, The Netherlands), APDM-Opal (APDM INC., Portland, OR, U.S.A.), and Shimmer-Shimmer3 (Shimmer Sensing, Dublin, Ireland)) and for three rotation rates (slow, medium, and fast motions) using the orientation provided by a multi-camera stereo-photogrammetric system (SP) as ground truth. For the sake of analysis generalizability, the SFAs performance is assessed under optimal and default parameters tuning. In the optimal tuning configuration, parameters are determined by minimizing the absolute orientation error with respect to the gold standard for each experimental scenario, allowing for the assessment of the filter performance under its best possible conditions. In addition, the errors obtained using the default parameter values as defined by the SFA proponents were also computed to highlight the impact of using non-tuned and generic parameter values for different experimental scenarios. Computation time of the different SFAs was also evaluated.

To the best of our knowledge, this study is the most comprehensive study evaluating a considerable number of SFAs under optimal filter parameter tuning condition and under various experimental scenarios.

## 2. Materials and Methods

### 2.1. Optimal Working Conditions

To work properly, each SFA requires the tuning of a variable number of parameters [32]. In the present context, optimal working conditions refer to the parameter values providing the lowest absolute average orientation error for a given experimental data recording (i.e., they are specialized for each dataset) and hence the best achievable performance (best case scenario). In other words, each parameter value of each SFA was optimally tuned on each of the nine experimental scenarios (three rotation rates for three commercial products). The selection of the optimal parameter values is performed relying on the gold standard orientation. This strategy is implemented exclusively for comparative purposes and may be replicated only with the aid of a reference orientation (e.g., SP system with sub-millimeter accuracy).

### 2.2. Selected Algorithms

A total of ten SFAs, including five complementary filters and five Kalman filters were selected among the most popular and performing ones: Mahony et al. 2008 [6] (MAH), Madgiwck et al., 2011 [9] (MAD), Sabatini 2011 [34] (SAB), Valenti et al., 2015 [2] (VAC), Ligorio and Sabatini 2015 [12] (LIG), Valenti et al., 2016 [13] (VAK), Seel et al., 2017 [16] (SEL), Guo et al., 2017 [17] (GUO), MATLAB complementary filter R2020a (MCF) which is the MathWorks implementation of VAC but with only two parameters, MATLAB Kalman filter R2020a (MKF), which is the MathWorks implementation of the filter by Roetenberg et al., 2005 [5,35]. The details for each SFA are reported in Table 1 including the total number of parameters exposed.

As stated in the introduction, all the SFAs are based on the angular velocity integration to obtain a first approximation of the orientation estimate. The differences are related to how the accelerometer and magnetometer measurements are used to compensate for the drift caused by the time integration of the angular velocity and to additional strategies designed to deal with the linear acceleration and the ferromagnetic disturbances. In the following, a short description of each filter is given.

MAD is a CF in which the accelerometer and the magnetometer measurements are fused by means of a gradient descent algorithm. For the magnetic readings, only the horizontal projection is used to correct the orientation. The fusion process is governed by a unique parameter. A low value of it gives more weight to the gyroscope measurements. MAH is a CF which considers the discrepancy between the measured Earth’s fixed vector (gravity and magnetic field) and their estimates obtained using the previous orientation. This discrepancy (called error) is then weighted by a parameter and subtracted from the gyroscope signal before its integration. As opposite to MAD the magnetic readings influence both the attitude and heading. In neither filter is it possible to weigh differently the accelerometer and the magnetometer contributions and no strategy is implemented to compensate for the linear acceleration or the magnetic disturbances. SEL is a CF with independent accelerometer-based inclination correction and magnetometer-based heading correction. The latter is purely horizontal, which ensures that magnetic disturbances cannot affect the inclination. The algorithm is parameterized via two correction constants for the inclination and heading disagreements, one optional bias estimation parameter and an adaptation factor that reduces the weight of the accelerometer readings during dynamic motions. VAC is a CF which employs the accelerometer readings to correct the inclination by comparing the actual and the observed gravity direction. The magnetometer readings are then projected onto the horizontal plane and the angle between the observed magnetic North and the estimated one is used to correct the heading. The two correction processes are governed by two independent gains. VAC implements a linear two-thresholds method to progressively reject the measurements whereas their magnitude exceeds the expected value (i.e., 9.81 m/s^2^ and the local magnetic norm, respectively). MCF is the implementation of VAC by MathWorks from Sensor Fusion and Tracking Toolbox.

VAK, LIG GUO, SAB, and MCF belong to the class of the KFs. As a general rule, the weight given to the information provided by each of the three sensors is governed by dedicated parameters. In particular, the higher is the value of these parameters the less the information provided is trusted. For this reason, they are called “inverse weight”. A typical feature of the KF is the possibility to track the disturbances with the “state-vector augmentation technique”. While on one hand it represents an advantage, on the other hand each quantity tracked in the state vector must be weighted with dedicated parameters to account for the uncertainty in their modelling and, above all, a large state vector dimension may result in observability problem (i.e., the information contained in the output variables is no longer sufficient to completely describe the system behavior).

VAK employs the same algebraic approach of VAC to correct the orientation, but as opposite to VAC no thresholds are used to reject linear accelerations or ferromagnetic disturbances which are instead employed in SAB. In addition, SAB allows the modelling of the ferromagnetic disturbances, seen as a time-variant bias superimposed to the magnetometer readings. LIG consists of two KFs which separately estimate the inclination and the heading, using the information provided by the gyroscope/accelerometer and gyroscope/magnetometer, independently. The two pieces of information are then merged by using an algebraic method. Linear accelerations and ferromagnetic disturbances are modelled as a first order Gauss-Markov model. GUO is a KF explicitly designed to perform fast. To this end, an algebraic approach which fuses the accelerometer and magnetometer measurements for the orientation correction is adopted and no additional strategy to filter out the linear accelerations and ferromagnetic disturbances is implemented. MKF is the MathWorks implementation (Sensor Fusion and Tracking Toolbox) of the filter originally proposed by Luinge et al. [4] and extended by Roetenberg et al. [5] which is also embedded in the Xsens software. Differently from the other four KFs described, MKF is an indirect KF, which means that it minimizes the uncertainty of the orientation error rather than of the absolute orientation (direct formulation). In this filter, the inclination and heading errors are separately computed by comparing the actual and the estimated directions of the gravity and global magnetic field using the information provided by gyroscope/accelerometer and gyroscope/magnetometer, respectively. These two orientation errors are then included in the state vector to be minimized. MKF augments its state vector with the gyroscope bias, acceleration errors (seen as the linear acceleration component in the accelerometer output) and the ferromagnetic disturbances.

For each SFA, the optimal tuning involved a heuristic space search of the two most important parameters (namely $p_{1}$ and $p_{2}$) when relevant. The decision to tune at most two parameters was a compromise between the search space dimension and the related computational time. As a general rule, considering that the gyroscope is the main source of information in a sensor fusion framework, the parameter related to the weight given to it was always tuned, when exposed by the SFA. All the remaining parameter values are set to default. In fact, it has to be said that the accelerometer and magnetometer related parameters should not be set based on the sensor noise only (i.e., electrical noise) because in the strict sense it is not representative of real problems affecting the two sensors: the linear accelerations and the ferromagnetic disturbances.

All the SFA codes have been made available on GitHub website (the link is reported in “Data Availability Statement” section). All the implementations are written in MATLAB code. All the details for each implementation are given in the code headers.

### 2.3. Experimental Setup

Considering that the performance of the SFAs could greatly vary due to the hardware characteristics, in this study, three pairs of commercial MIMUs were considered: Xsens-MTx, APDM-Opal, Shimmer-Shimmer3. The specifications for each model are reported in Table A3, Table A4 and Table A5 of Appendix B. A wooden board was used and the three MIMU pairs were aligned as depicted in Figure 1. A T-square was employed to draw the lines which ensured the accurate positioning of the MIMUs and the markers. The orthogonal tolerance of the instrument is reflected in an alignment error lower than 0.2 deg. The MIMUs were positioned at a relative distance of 50 mm. A total of eight reflective spherical markers (diameter equal to 14 mm, minimum inter-distance of 85 mm) were located on the board to provide the orientation reference whose trajectories were acquired by 12 infrared SP cameras (Vicon T20, VICON, Yarnton, UK; software: Nexus 2.7). The three central markers were used to define the SP Local Coordinate System (LCS) aligned with the MIMUs LCS. The additional five markers were exploited to strengthen the orientation estimation by using the singular value decomposition (SVD) technique [36]. The board was placed over an aluminum tripod (to limit the entity of the ferromagnetic disturbances) above a force platform integrated with the SP and used to synchronize the two systems by means of mechanical shocks.

### 2.4. Experimental Protocol

Before starting the experiments, a ten-minute instrument warm-up was performed to limit the temperature effects on the sensor readings. A first measure consisted of one-minute static acquisition with the board lying horizontally on the tripod above the force platform. This recording was used to compute the gyroscope bias which was subtracted from the angular velocities’ readings. Afterwards, a dynamic recording was executed to manually orient the board by covering the three rotational degrees of freedom. The operator held the board at both ends and performed both single-axis and multiaxial rotations (the acquisitions have been video-recorded and made available). At the end, the board was placed back on the tripod for another minute. To allow the identification of the synchronization points, two knocks were given to the board at the beginning and at the end of the recording. The described protocol was repeated at three different angular rate conditions. The RMS of the angular velocity of each recording was assessed during the post processing and amounted at 120 deg/s for a total of 70 s (slow), 260 deg/s for a total of 45 s (medium), and to 380 deg/s for a total of 30 s (fast). The temperature was kept constant at about 20 °C.

All the acquisitions were conducted in a volume of approximately 1 m^3^ at a distance greater than 1 m from the floor. For this reason, the ferromagnetic disturbances could be neglected as also observed in the post processing by observing the almost constant magnetometer norm (the maximum difference was limited to 1 µT).

The sampling frequency of the MIMU systems amounted to 100 Hz for Xsens (MT Manager Version 1.7) and Shimmer (Consensys v.1.5.0), and to 128 Hz for OPAL (Motion Studio Version 1.0.0.201712300). The calibrated data provided by each software were used in this work. Optical data were recorded at 100 Hz. All the MIMU synchronized data and the SP orientation (computed as described in Section 2.5.1) together with the videos recorded during the experiment have been uploaded as stated in the Data Availability Statement section at the end of the manuscript [37].

### 2.5. Data Processing

#### 2.5.1. SP Data Pre-Processing and Synchronization with MIMUs Signals

Data processing was entirely carried out in MATLAB R2020a (The MathWorks Inc., Natick, MA, USA), except for the optical data which were first processed in Nexus 2.7 following the suggestion by Bergamini et al. [25]. The synchronization was performed in two steps: firstly, all data were delimited by means of the two force and acceleration peaks recorded by the vertical axes of the SP force plate and of the MIMU accelerometer, respectively. Afterward, a resampling of all the signals at 100 Hz was executed. To refine the synchronization, all the MIMU data were aligned with those of SP by cross-correlating the angular velocity recorded by each gyroscope and that estimated by the three central markers as described in [38]. The LCS orientation was estimated with respect the GCS of SP by means of SVD-based technique. The resulting gold standard orientation ($q_{SP_{G}}$) was expressed using quaternion. After trigonometry considerations (taking into account the cluster size of the three central markers and that the marker position errors amounts to about 0.1 mm [31,39]), it can be assumed that the errors affecting the gold-standard orientation are limited to 0.5 deg.

#### 2.5.2. Orientation Estimation and Error Computation under Optimal Conditions

The procedure used for the orientation estimation for each SFA is detailed below and reported in the pseudocode (Algorithm 1). In the following, the procedure used to obtain the set of orientations for a given SFA for each of the nine experimental scenarios is briefly described. Quantities highlighted in bold are intended to be vectors or matrices.

An algebraic quaternion, obtained with only the accelerometer and magnetometer measurements [2], was used to initialize the orientation of each MIMU to reduce the convergence time. For each MIMU (A and B) the absolute orientation ($q_{A_{G}}$ and $q_{B_{G}}$) was computed separately for every combination of the values of the two parameters (stored in $p_{1vec}$ and $p_{2vec}$, respectively) from 0 to $upper_{1}$ and from 0 to $upper_{2}$, respectively. The only exception is represented by a_th2_ of VAC whose lower limit was set to the value of the first accelerometer threshold (a lower value would be meaningless). The values for $upper_{1}$ and $upper_{2}$ were chosen to be large enough to ensure the exploration of all the relevant search space. In other words, errors obtained for values of $p_{1}$ and $p_{2}$ set to $upper_{1}$ and $upper_{2}$, respectively, are large. Figures in Appendix C display the values chosen for $upper_{1}$ and $upper_{2}$ for each SFA. The number of points for each parameter interval (i.e., the length of the $p_{1vec}$ and $p_{2vec}$ vectors) was different for each algorithm and it is a trade-off between the search space dimension and the computational cost; on average about 360 solutions (i.e., length ($p_{1vec}$) × length ($p_{2vec}$)) were explored for each SFA. Since the GCSs of the MIMU and SP were not aligned on the horizontal plane, to enable a meaningful comparison between the orientation obtained for the two systems, it was necessary to refer the latter to a common GCS. To this end, it was possible to benefit from the accurate alignment of the LCS of each system. Therefore, $q_{A_{G}}$, $q_{B_{G}}$, and $q_{SP_{G}}$ were separately referred to their initial frame to obtain $q_{A}$, $q_{B}$, and $q_{SP}$, respectively, as follows (the ⊗ and ^*^ operators represent the product and complex conjugate operator in the quaternion algebra, respectively):(1)$$\begin{array}{c} q_{A}=\;q_{A_{G}}{\left(1\right)}^{*}\;⊗q_{A_{G}},\\ q_{B}=\;q_{B_{G}}{\left(1\right)}^{\;*}\;⊗q_{B_{G}},\\ q_{\mathrm{SP}}=\;q_{SP_{G}}\left(1\right){\;}^{*}\;⊗q_{SP_{G}}. \end{array}$$

The absolute orientation errors $\Delta q_{abs\;A}$ and $\Delta q_{abs\;B}$ were computed in the quaternion form as follows:(2)$$\begin{array}{c} \Delta q_{abs\;A}=\;q_{A}{}^{*}\;⊗q_{SP},\\ \Delta q_{abs\;B}=\;q_{B}{}^{*}\;⊗q_{SP}. \end{array}$$

To obtain a compact representation of the errors and the relative difference, the absolute rotation angle was computed from the scalar component of each quaternion Δqabs A and Δqabs B to obtain Δθabs A and Δθabs B, respectively. Then, the two absolute error values Δθabs A and Δθabs B were averaged to obtain Δθabs. Lastly, the RMS value of Δθabs was computed only during the dynamic portions of the recording to obtain ep1, p2. This procedure was repeated for each combination of p1vec and p2vec to populate the e matrix which contains the absolute errors which were rounded to an accuracy of 0.1 deg.
**Algorithm 1.** Pseudocode to detail the orientation estimation process for each SFA. **for** each pair of MIMUs (Xsens, APDM, and Shimmer)  **for** each angular rate condition (slow, medium, fast)remove the static bias for each gyroscopecompute the starting orientation for each MIMUinitialize the matrix e (#rows = length(p1), #columns = length(p2)) **for** each value p2 belonging to p2vec between [0, upper2] **for** each value p1 belonging to p1vec between [0, upper1]compute the absolute orientation of each MIMU separately with the SFA under analysis to obtain qAG and qBGrefer qAG and qBG to the starting orientation to obtain qA and qB, as done in (1)compute the absolute orientation error of qA and qB separately using the gold standard qSP to obtain Δqabs A and Δqabs B, as done in (2)convert Δqabs A and Δqabs B into angular rotation errors to obtain Δθabs A and Δθabs Bcompute the average value between the two absolute errors to obtain Δθabscompute the RMS of Δθabs considering only the dynamic parts of the recording to obtain ep1, p2add ep1, p2 to the matrix e **end** **end**find the optimal region of (p1vec,p2vec) which correspond to the range of e which includes its minimum (eopt) + 0.5 deg to obtain popt_1 and popt_2find the value of e which correspond to the default parameter values to obtain edef **end****end**

### 2.6. Data Analysis

The evaluation of the algorithms’ performance followed the steps described in the following. The first step consisted in identifying the optimal regions and the corresponding errors for each SFA and for each experimental scenario (i.e., 90 errors in total, 10 SFAs × 3 rotations rates × 3 commercial products). The absolute orientation errors were also computed for each SFA and for each scenario using the default parameter values listed in Table 1 (i.e., 90 errors) which were provided by the authors in their papers or in the original implementations of their SFAs. Then, the influence of the following factors on the absolute orientation accuracy was analyzed:SFA analytical formulationrotation rate magnitudedifferent commercial products.

To this extent, a statistical analysis was performed by aggregating data according to the influencing factor under inspection.

Finally, an analysis of the computation time of each SFA was performed by measuring the amount of time needed by each SFA to perform a single orientation update iteration (i.e., a single time step).

#### 2.6.1. Identification of the Optimal Regions and the Corresponding Errors

The optimal region, for each scenario, is defined as the combination of $p_{1}$ and $p_{2}$ which corresponds to the minimum of absolute orientation error.

For each scenario, the following quantities were determined:
Minimum absolute orientation error which corresponds to the selection of the optimal parameter values: $e_{opt}=\min\left(e\left(p_{1},p_{2}\right)\right)$, where e is the matrix of the average errors between the two MIMUs of dimensions equal to [length($p_{1}$), length($p_{2}$)].Optimal parameter region is defined as the range of parameter values for which the relevant orientation errors are equal to the minimum error plus 0.5 deg (i.e., the SP uncertainty band, as stated in Section 2.5.1). These regions are defined as: $\left\{p_{opt_{1}},p_{opt_{2}}\}=\{\left(p_{1},p_{2}\right)\;|\;e\;\leq e_{opt}+0.5\;deg\right\}$. An example of optimal region is illustrated in Figure 2 for the VAK filter. When only one parameter was tuned (MAD, VAC, GUO, MKF) e was a vector and the optimal region degenerated into a 1D interval.

#### 2.6.2. Identification of the Default Errors

The absolute error corresponding to the default values of $\left(p_{1_{DEF}},p_{2_{DEF}}\right)$ was obtained as: $e_{DEF}=\;e\left(p_{1\_DEF},p_{2\_DEF}\right)$.

#### 2.6.3. Statistical Analysis to Evaluate the Influence of the SFAs and of the Experimental Factors

To evaluate the influence of the different factors on the 90 values of $e_{opt}$ were aggregated as explained in the following Table 2:

For each influencing factor, the normality of each distribution was tested with a Shapiro-Wilk’s test suitable for small sample size. Since it turned out that the distributions were not normal, the nonparametric Friedman’s test was applied to assess whether significant differences existed among the error distributions. When the null hypothesis was rejected, post-hoc multiple comparison tests were applied to perform pairwise comparisons. In particular, to test the SFA influence, the Tukey’s honest significant difference criterion was used since it is less strict than Bonferroni having 10 distributions [40]. Bonferroni’s correction was instead used to perform pairwise comparisons when testing the rotation rate and commercial product effects.

#### 2.6.4. Computation Time of the Different SFAs

The average execution time for a single iteration of each SFA is measured for an Intel^®^ Core™ i7-10510U CPU @ 1.80 GHz (Intel ©, Santa Clara, CA, USA)—Microsoft™ Windows 10 (Microsoft ©, Redmond, WA, USA) when processing a dataset of 25386 samples without executing any other programs.

## 3. Results

### 3.1. Optimal and Default Errors

The minimum absolute error ($e_{opt}$) and the errors corresponding to default parameter values ($e_{def}$) are reported in Table 3 for each SFA and for each experimental scenario.

### 3.2. Optimal Regions

The optimal regions for each experimental scenario for each SFA are reported in the figures in Appendix C for sake of completeness. Figure 2 provides an example of the optimal regions determined for each of the nine experimental scenarios in the case of VAK. In more detail, the optimal region identified by $p_{opt\_1}$ and $p_{opt\_2}$ is represented with a different color for each scenario.

### 3.3. Influence of the Experimental Factors on the Absolute Accuracy

#### 3.3.1. Influence of the Specific SFA (3 Rotation Rates × 3 Commercial Products)

Mean ± STD of the *e_opt_* values obtained by each SFA are listed in ascending order in Table 4.

As evident from Table 3, in many conditions the use of the default parameters (listed in Table 1) does not guarantee for low values of $e_{def}$ for each SFA. For this reason, the analyses performed in the following sections to investigate the influence of the rotation rate and the commercial product were limited to $e_{opt}$ distributions (it would be meaningless for those of $e_{def}$).

The Shapiro-Wilk test, (α = 0.05) revealed that not all the $e_{opt}$ distributions were normal (*p* < 0.05). The small *p*-value (0.0035) resulting from the Friedman’s test cast doubts on the validity of the null hypothesis. A multiple comparison test with Tukey’s correction (α = 0.05) revealed that no statistically significant differences existed among the 10 SFAs under optimal working conditions.

#### 3.3.2. Influence of the Rotation Rate (10 SFA × 3 Commercial Products)

Optimal distributions are represented in Figure 3 for slow, medium, and fast rotation rates. In addition, the mean ± STD errors for each rotation rate scenario are reported in Table 5.

The Shapiro-Wilk test, (α = 0.05) revealed that not all distributions were normal (*p* < 0.05). The small *p*-value (<1 × 10^−9^) resulting from the Friedman’s test cast doubts on the validity of the null hypothesis. A multiple comparison test with Bonferroni’s correction (α = 0.05) revealed a statistically significant difference among the three distributions (Table 6).

#### 3.3.3. Influence of the Commercial Product (10 SFA × 3 Rotation Rates)

Noise description for each sensor of each MIMU and gyroscope bias were reported in Table A4 and Table A5, respectively (Appendix B) [41].

Optimal distributions are represented in Figure 4 for Xsens, APDM, and Shimmer commercial products. In addition, the mean ± STD errors for each commercial product scenario are reported in Table 7.

The Shapiro-Wilk test, (α = 0.05) revealed that not all the distributions were normal (*p* < 0.01). The small *p*-value (<1 × 10^−7^) resulting from the Friedman’s test cast doubts on the validity of the null hypothesis. A multiple comparison test with Bonferroni’s correction (α = 0.05) revealed a statistically significant difference among the three distributions (Table 8).

### 3.4. Computation Time of the Different SFAs

In Figure 5, the average execution time of each SFA is reported for a single iteration.

## 4. Discussion

### 4.1. The Importance of Properly Tuning Each SFA

Results from the present study have confirmed that the selection of appropriate parameter values plays a fundamental role in determining the level of orientation accuracy and that parameters must be tuned differently based on the experimental scenario [15,27,31], thus enabling the best possible performance.

We found that each SFA exhibits the optimal performance only for a limited interval of parameter values. If the parameter values are optimized for a specific experimental scenario, the same values can lead to large errors when varying the experimental conditions; the only exception is represented by the MKF with a common intersection between $\sigma_{gyr}^{2}$ = [0.0125, 0.0275] (rad/s)^2^. This evidence can be also graphically observed in the figures in Appendix C, in fact for all algorithms but MKF there is not a common intersection among the optimal regions when varying the experimental scenario. The above-mentioned figures also prove that for some SFAs (e.g., SAB), the specific tuning for each experimental scenario is particularly critical since the overlapping among the optimal regions is very limited. At the same time the errors obtained using the default parameter values ($e_{def}$) highlight the inadequacy to estimate the absolute orientation with the same parameter values for a given SFA under different experimental scenarios. These findings provide a further justification of the different level of accuracy reported for the different SFAs in previous studies which entailed the comparison among filters optimally tuned and filters fed with the default or non-optimal parameters (e.g., [14,16]). It is clear that any comparison carried out without a common strategy to tune the SFA parameter values would be lacking generality. In fact, for some algorithms $e_{def}$ can greater than 100 deg (see Table 3). It is worth pointing out that the optimal errors reported in Table 3 can be considered the lower bound for those SFAs under similar experimental conditions. Lower errors can be only obtained under less challenging scenarios and/or using higher performing MIMUs.

As already mentioned, the optimal parameter tuning requires the orientation reference to be available (e.g., a SP system). Since the MIMUs are conceived to be used also outside the laboratory, the proposed approach is not always feasible and there is the need to search for alternative strategies for the selection of reasonable parameter values without using any orientation reference. To the best of our knowledge, the procedure described in [31] is the only one which meets this requirement by exploiting the fact that two MIMUs aligned on a rigid body must have a null orientation difference during any movement.

### 4.2. Influence of the SFA and of the Experimental Factors on the Absolute Accuracy

The influence of the specific SFA and the filtering class (CF or KF) on the orientation accuracy has been widely investigated by several authors (e.g., [7,25]). We found, over SFAs, errors ranging from 3.8 deg to a maximum of 7.1 deg, but no statistically significant differences. This suggests that, based on the present study, it is not possible to identify the best performing SFA and that a proper fine tuning of the parameter values can be the key point to obtain a reasonable absolute accuracy, regardless of the filtering class or the total number of parameters exposed by the SFAs. In fact, it is possible to observe from Table 1 and Table 3 that a larger number of parameters does not necessarily guarantee for a better accuracy with respect to a SFA with only one parameter to be set. This can be explained by the fact that, ideally, the several sources of errors can be better modelled by filters with a large number of parameters. On the other hand, their tuning is more difficult since the final orientation estimate is strongly influence by the mutual influence of the parameters. Among the ten tested SFAs, LIG exhibited the lowest average errors while VAC and MCF the highest being the average differences under 0.5 deg (this is expected since they are the implementations of the same filter described in [2]). It has to be said that this experimental design was not conceived to enhance small differences across the performance of ten algorithms due to the weak statistical power. If of interest, several repetitions for each experimental scenario would need to be collected.

Results showed that even under optimal parameter tuning orientation accuracy is strongly influenced by the experimental factors considered.

The influence of the rotation rate magnitude has been recognized by previous works ([26,27,33,42]) in which accuracy worsening was observed when the rotation rate increased, however few SFAs were tested. In general, higher values of rotation rate are associated with higher linear accelerations (except when the MIMU is coincident with the center of rotation), which are directly reflected on the specific force recorded by the accelerometer. Since the accelerometer aids the sensor fusion process by providing the gravity direction information to compensate for the inclination drift, when the gravity recording is corrupted by high values of linear acceleration then the accelerometer contribution becomes detrimental. Many algorithms cope with this problem by rejecting the accelerometer information when the magnitude of its measures overcome a certain threshold. However, as highlighted by Fan et al., in [18], despite the simplicity of this strategy that can be adopted by both CF and KF classes, the main drawback is the choice of the threshold value and the resulting orientation instability for accelerometer values close to the threshold. Observing the results in Table 5, the effect of the rotation rate is confirmed also for all the SFAs analyzed: errors obtained at the fast rotation rate are worse than those at slow rotation rate and the three distributions statistically differ across all of them (Figure 3). In particular, the performance worsens of 3.8 ± 2.1 deg on average. The minimum worsening amounts to 0.2 deg for SAB-Xsens and the maximum to 9.7 deg for GUO-Shimmer.

Finally, it is known that the hardware components embedded in the commercial products, although sharing similar specifications (measurement ranges, sensitivity, resolution, etc.) exhibit different noise levels both in terms of STD and offset and bias instability, which in turns influence the estimation accuracy [26,33,42]. In particular, the most critical factor when estimating the orientation is the slow-varying bias affecting the gyroscopes. Table A3 and Table A4 reported in the Appendix B show the measured noise levels for different commercial products and the difference between the gyroscope bias computed at the beginning and at the end of the recordings. The differences between the noise levels of two MIMUs of the same commercial product are evident. Obviously, when considering different commercial products, the differences are even higher. Moreover, the bias of the gyroscopes changed during the same recordings of a few minutes. The problem of a slow varying bias is one of the major problems to address when estimating orientation since the angular velocity is the main source of information in a sensor fusion process. This may partially explain the significant difference found in Table 8 and shown in Figure 4 between Xsens and APDM for which the bias changes are up to two orders of magnitude higher. However, the exact mechanism with which the noise level is reflected on the absolute accuracy is still not completely clear due to the high number of variables involved in the sensor fusion process. It is worth highlighting, as a limitation, that the different locations of the MIMUs on the board, which lead to different acceleration magnitudes, can have a minor and limited influence on the results. In fact, as shown in Figure 4, the medians of the APDM and Shimmer distributions are extremely similar. Overall, Xsens showed the smallest average errors of 3.5 deg, while Shimmer the highest of 6.5 deg.

### 4.3. Computation Time of the SFAs

Depending on the specific application, the computation time required by each SFA to compute the orientation may be crucial. In those applications providing feedback to the patient such as the tele-rehabilitation and neuroprosthesis systems, the “near” real-time estimation is a fundamental requirement [43].

It is observed that most tested CFs are faster than the KFs. The two exceptions are GUO (a KF explicitly designed by the authors to perform fast), which resulted to be the second fastest and MCF (a CF), which is the MATLAB implementation of Valenti 2015 (VAC). MCF is more than 32 times slower than VAC, despite the mathematical formulation being the same. This aspect has been analyzed through the MATLAB profiler and it emerged that the quaternion library used by MATLAB (more specifically the creation of a quaternion object) is the limiting factor. The same applies to MKF (the MathWorks implementation of [5] embedded in the “Sensor Fusion and Tracking Toolbox”) which is the slowest KF. The higher computation time of Kalman filters can be due to the matrix operations involved in the mathematical implementation including multiplications and inversions. Finally, a monotonic relationship between the dimension of the state vector and the time required to complete an iteration can be observed. This is justified by the fact that an increase in the state vector dimension involves matrices of higher dimensions (e.g., state transition, process and measurement covariance matrices, etc.) with a consequent increase of the computational burden. It has to be said that not all the MATLAB implementations were optimized by the authors to perform fast. The same implementations written in another programming languages such as C/C++ could be executed much faster.

## 5. Conclusions

In this study the two most influencing parameters were identified and optimally tuned for each SFA, thus obtaining the corresponding errors, which are indicative of the best possible performance under each tested experimental scenario. Moreover, as suggested by recent literature [15,27,31,44,45], this study confirms that the selection of the value of each parameter is crucial to obtain a satisfying performance of each SFA despite its filtering class or mathematical formulation. The use of fixed parameter values may be not suitable for every scenario since, in general, the optimal value regions do not intersect. The authors stress the importance of comparing the performance of the different SFAs only when all of them are run under the same working conditions and suggest all future works to follow this recommendation. This study also analyzes how accuracy is influenced by the SFA, the rotation rate, and the commercial product.

An important finding is that all methods exhibited errors within a range of 3.3 deg (from 3.8 deg to 7.1 deg). Therefore, it is difficult to rank the ten implemented SFAs, and it is not possible to identify the best performing SFA since no statistically significant differences were found.

Errors at high rotation rate are statistically different from those obtained at low rotation rate. On average, the lowest errors associated with a slow movement are expected to be about 3.5 deg and they could increase up to 7.3 deg when the rotation rate reaches an RMS value of about 380 deg/s. The errors are also influenced by the model of the commercial product. Xsens provided the lowest average errors (3.5 deg) and statistically different with respect to APDM (6.0 deg), and Shimmer (6.5 deg).

Finally, complementary filters were found to be faster than Kalman filters. This can be substantiated by the mathematical implementation of the filters. Two exceptions were observed: GUO (a KF explicitly designed by the authors to perform fast) and MCF (the MATLAB CF). In addition, an increase of the state vector dimension is monotonically reflected on the time required for each iteration to estimate the orientation.

In conclusion, this study shows the importance of equally tuning the parameter values for each SFA in order to enable a meaningful comparison among the different algorithms. In addition, performances are strongly influenced by the experimental conditions. It has to be pointed out that while this contribution, for the best of our knowledge, is the most extensive so far (10 SFAs × 3 motion intensities × 3 commercial products) there are many relevant aspects that are also worth considering when evaluating the accuracy of SFAs, such as the effect of translations, long uninterrupted motion phases, and the influence of magnetic disturbances. The proposal of a complete and standard benchmark to test the SFAs under different experimental scenarios would be beneficial for any future comparison. As pointed out by Nazarahari and Rouhani [29] there is still the lack of a shared protocol and the movement analysis community should move in this direction.

## Figures and Tables

**Figure 1 sensors-21-02543-f001:**
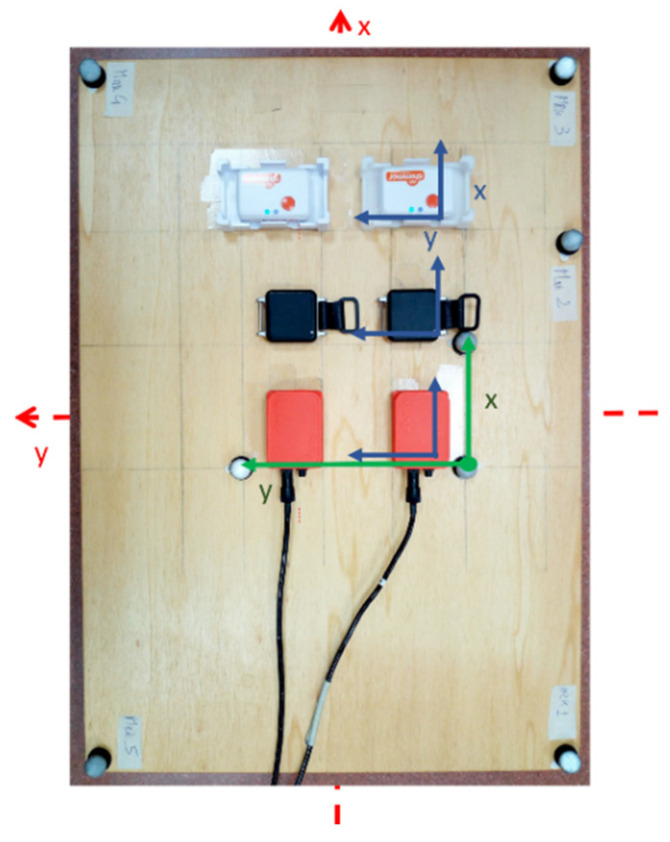
Board equipped with six magneto-inertial measurement units (MIMUs) (relevant local coordinate system (LCS) in blue) and the eight reflective markers. The three central markers were used to define the stereo-photogrammetric (SP) system LCS (in green). Board axes (in red) are coincident with MIMUs and SP system LCSs. Reprinted with permission from ref. [31]. Copyright 2020 IEEE.

**Figure 2 sensors-21-02543-f002:**
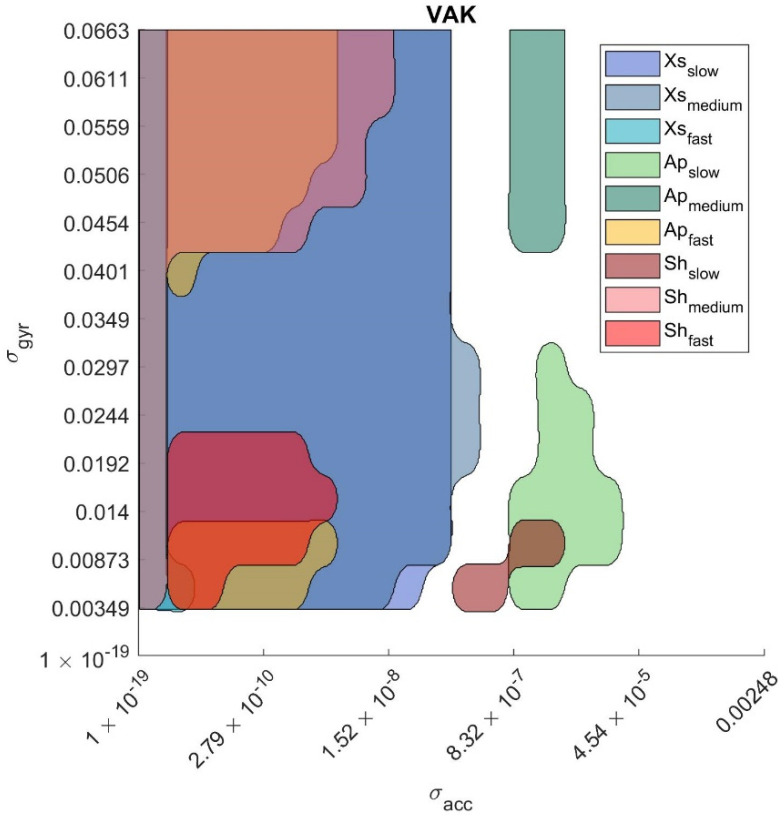
Optimal regions (one for each experimental scenario) for Valenti et al., 2016 (VAK). σ_acc_ values are exponentially spaced.

**Figure 3 sensors-21-02543-f003:**
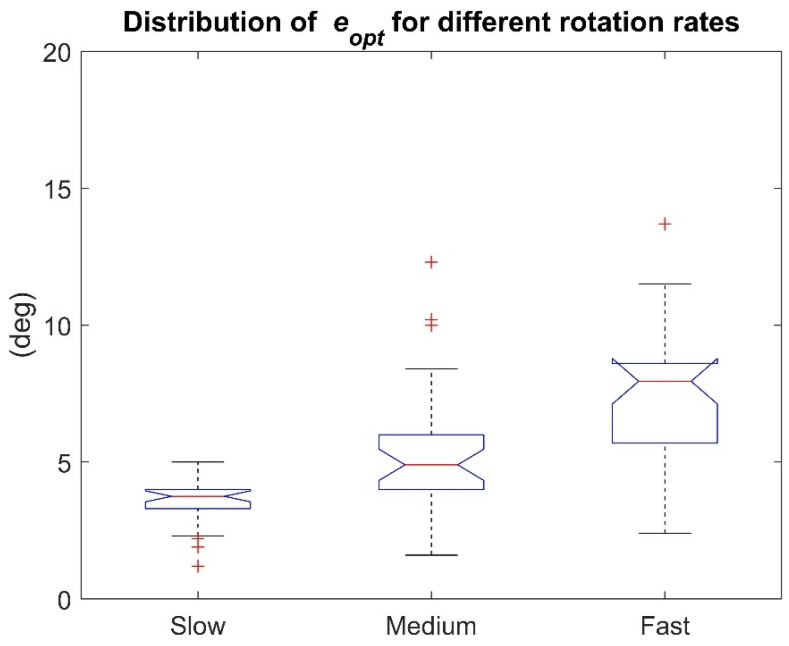
Rotation rate effect: slow, medium, and fast $e_{opt}$ distributions (10 SFAs × 3 commercial products). It is possible to assess that errors obtained at the fast rotation rate are worse than those at slow rotation rate and the three distributions statistically differ across all of them.

**Figure 4 sensors-21-02543-f004:**
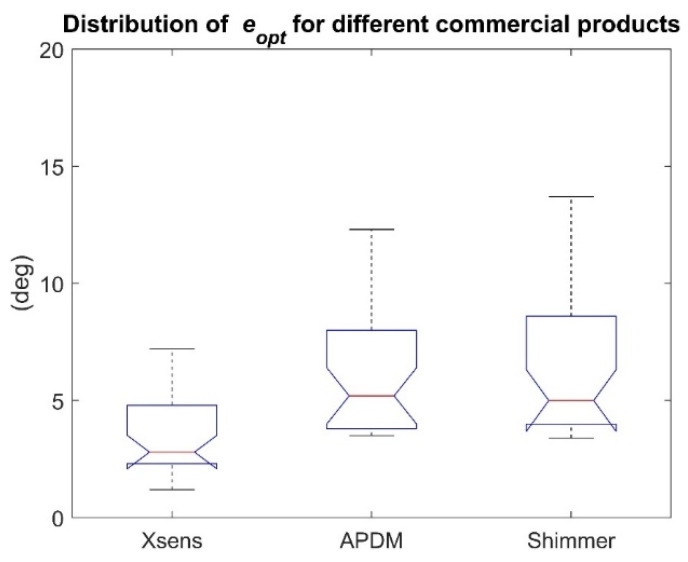
Commercial product effect: Xsens, APDM, and Shimmer $e_{opt}$ distributions (10 SFAs × 3 rotation rates). It is possible to assess that APDM and Shimmer distributions statistically differ from that of Xsens.

**Figure 5 sensors-21-02543-f005:**
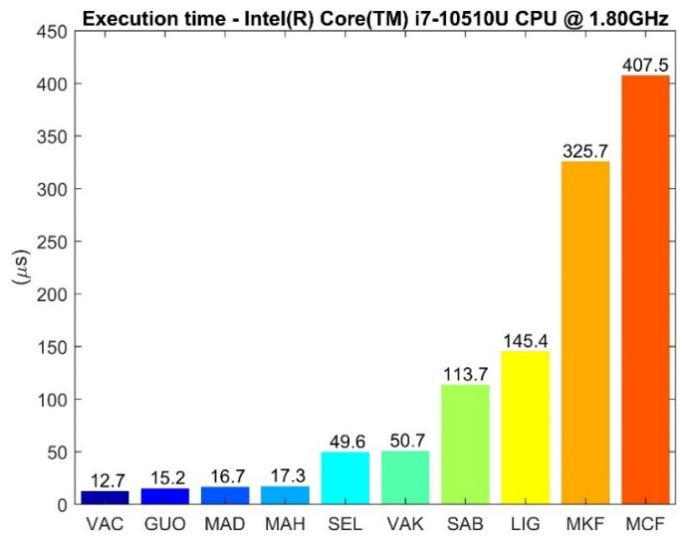
Execution time needed for a single update step by each SFA.

**Table 1 sensors-21-02543-t001:** Details of each Sensor Fusion Algorithm (SFA) considered. The # Params column reports the total number of parameters of each SFA. The *p_1_* and *p_2_* report the description of the parameter tuned to detect the optimal values. a.u. = arbitrary units.

**CF**	**# Params**	$p_{1}$	**Default**		$p_{2}$	**Default**	
MAH	2	k_p_—inverse gyroscope weight	1	rad/s	k_i_—weight for online bias estimation	0.3	rad/s
MAD	1	β—inverse gyroscope weight	0.1	rad/s	/	/	
VAC	9	g_mag_—magnetometer weight	0.01	a.u.	a_th2_—threshold for accelerometer vector selection	0.2	a.u.
SEL	4	τ_acc_—accelerometer time constant	1	s	τ_mag_—magnetometer time constant	3	s
MCF	2	g_mag_—magnetometer weight	0.01	a.u.	/	/	
**KF**	**# Params**	$p_{1}$	**Default**		$p_{2}$	**Default**	
SAB	6	σ_gyr_—inverse gyroscope weight	0.007	rad/s	a_th_—threshold for accelerometer vector selection	40	mg
LIG	6	σ_gyr_—inverse gyroscope weight	1	rad/s	c_b_—Gauss-Markov parameter of the prediction model to set the variance of external acceleration and ferromagnetic disturbances	1	a.u.
VAK	3	σ_gyr_—inverse gyroscope weight	0.004	rad/s	σ_acc_—inverse accelerometer weight	0.014	m/s^2^
GUO	3	σ_gyr_—inverse gyroscope weight	0.001	rad/s	/	/	
MKF	8	σ^2^_gyr_—inverse gyroscope weight	9.14 × 10^−5^	(rad/s)^2^	/	/	

**Table 2 sensors-21-02543-t002:** Statistical analysis plan to evaluate the influence of SFAs, rotation rate, and commercial product on the errors.

Influencing Factor	Number of Distributions	Number of Values for Each Distribution
SFA	10 (one for each SFA)	9 (=3 rotation rates × 3 commercial products)
Rotation rate	3 (one for each rotation rate)	30 (=10 SFAs × 3 commercial products)
Commercial product	3 (one for each commercial product)	30 (=10 SFAs × 3 rotation rates)

**Table 3 sensors-21-02543-t003:** The Optimal Errors Are Reported with the Absolute Errors Obtained Using the Default Parameter Values.

		CF	$e_{opt}$	$e_{def}$	KF	$e_{opt}$	$e_{def}$
Xsens	Slow	MAH	2.5	4.2	SAB	2.2	67.9
	Medium		2.4	11.9		2.1	96.6
	Fast		4.0	13.0		2.4	53.9
APDM	Slow		3.8	3.9		5.0	77.5
	Medium		4.8	17.7		5.7	62.6
	Fast		8.2	12.3		8.3	9.9
Shimmer	Slow		3.4	5.9		4.5	71.1
	Medium		4.6	38.2		4.9	14.5
	Fast		7.6	17.0		8.5	30.0
Xsens	Slow	MAD	2.7	4.7	LIG	1.9	3.7
	Medium		2.5	5.2		2.0	3.9
	Fast		4.0	6.8		2.9	4.8
APDM	Slow		3.8	4.1		3.6	3.6
	Medium		4.6	4.6		4.9	5.0
	Fast		8.1	8.2		4.6	4.6
Shimmer	Slow		3.9	4.3		4.4	4.4
	Medium		4.9	5.2		4.0	4.2
	Fast		8.8	8.9		6.3	6.5
Xsens	Slow	VAC	4.0	4.1	VAK	1.2	22.3
	Medium		5.0	5.9		1.6	21.4
	Fast		7.2	10.0		2.5	72.8
APDM	Slow		3.5	3.6		3.6	29.6
	Medium		6.1	11.8		6.0	30.4
	Fast		8.3	15.1		9.2	81.9
Shimmer	Slow		3.8	3.8		4.0	32.6
	Medium		10.2	19.2		4.4	48.8
	Fast		11.5	23.6		8.2	100.1
Xsens	Slow	SEL	3.1	4.0	GUO	2.3	3.7
	Medium		2.5	4.6		2.3	4.9
	Fast		5.1	6.7		5.7	10.6
APDM	Slow		3.7	3.8		4.2	4.5
	Medium		7.1	7.3		5.1	5.3
	Fast		8.0	8.8		9.4	12.0
Shimmer	Slow		3.4	3.5		4.0	4.0
	Medium		5.0	8.4		5.1	5.7
	Fast		9.4	11.8		13.7	16.7
Xsens	Slow	MCF	3.3	4.5	MKF	4.2	4.9
	Medium		6.1	6.2		4.8	8.7
	Fast		6.6	7.8		6.7	10.9
APDM	Slow		3.8	4.2		3.6	4.8
	Medium		12.3	12.3		5.3	14.3
	Fast		7.9	9.3		7.2	10.7
Shimmer	Slow		5.0	5.2		3.9	5.8
	Medium		10.0	10.1		8.4	45.2
	Fast		8.6	12.0		9.9	19.0

All the unit are in degrees.

**Table 4 sensors-21-02543-t004:** The Mean ± STD of the Errors for Each SFA for Optimal Parameter Values Selection.

	LIG	VAK	MAH	MAD	SAB	SEL	GUO	MKF	VAC	MCF
$e_{opt}$	3.8 ± 1.4	4.5 ± 2.8	4.6 ± 2.1	4.8 ± 2.2	4.8 ± 2.4	5.3 ± 2.4	5.8 ± 3.7	6.0 ± 2.2	6.6 ± 2.9	7.1 ± 2.9

All units are in degrees.

**Table 5 sensors-21-02543-t005:** Mean ± STD Errors for Each Rotation Rate Scenario.

(deg)	Slow	Medium	Fast
$e_{opt}$	3.5 ± 0.9	5.2 ± 2.5	7.3 ± 2.6

**Table 6 sensors-21-02543-t006:** Results of Friedman’s Test with Bonferroni’s Correction to Investigate the Differences among the Three Rotation Rate Conditions.

Scenario	Optimal Conditions
Slow vs. fast	Significantly different (*p* < 1× 10^−4^)
Slow vs. medium	Significantly different (*p* < 1× 10^−3^)
Fast vs. medium	Significantly different (*p* = 0.013)

**Table 7 sensors-21-02543-t007:** Mean ± STD Errors for Each Commercial Product Scenario.

(deg)	Xsens-MTx	APDM-Opal	Shimmer-Shimmer 3
$e_{opt}$	3.5 ± 1.7	6.0 ± 2.3	6.5 ± 2.8

**Table 8 sensors-21-02543-t008:** Results of Friedman’s Test with Bonferroni’s Correction to Investigate the Differences among the Three Commercial Product Conditions.

Scenario	Optimal Conditions
Xsens vs. APDM	Significantly different (*p* < 1× 10^−5^)
Xsens vs. Shimmer	Significantly different (*p* < 1× 10^−6^)
APDM vs. Shimmer	Not significantly different (*p* = 1)

## Data Availability

The sensor fusion algorithm together with the optimization codes are available as MATLAB functions and scripts at https://github.com/marcocaruso/sensor_fusion_algorithm_codes (accessed on 27 March 2021). The complete dataset is available on both IEEE DataPort at http://dx.doi.org/10.21227/b23b-rx94 (last updated on 9 July 2020) [37] and also at https://github.com/marcocaruso/mimu_optical_dataset_caruso_sassari (accessed on 27 March 2021).

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
