# Peer review of "Analysis of the Accuracy of Ten Algorithms for Orientation Estimation Using Inertial and Magnetic Sensing under Optimal Conditions: One Size Does Not Fit All"

_sensors, 2021, doi:10.3390/s21072543_

Round 1

Reviewer 1 Report

The authors present an experimental evaluation of the accuracy of ten algorithms for fusing magnetic and inertial sensor data to determine orientation in human movement analysis. The algorithms include five complementary filters and five Kalman filters. The comparisons are performed over three different rotation rates and using three different commercial off-the-shelf IMU solutions under identical experimental conditions. As such, the paper provides a valuable assessment on an even playing field for these methods. Contrary to many past studies performing such comparisons, tuning is performed on up to two parameters for each algorithm in order to achieve the optimal performance under the experimental conditions. It is found that rotation rate and IMU product have a significant effect on the orientation accuracy when the algorithms are tuned optimally, but no significant effect was found between algorithms. The computation time for each algorithm is also analyzed. The authors also conclude that parameter tuning is key to obtaining reasonable accuracy for any algorithm.

Overall, the work presented is a valuable contribution to the literature and is largely technically accurate. My biggest concern with the paper is the procedure used to tune the algorithms. In particular, the algorithms were tuned on the same datasets used to evaluate them, which presents the potential for overfitting. While this likely does not negate the authors’ conclusions, the magnitude of improvement from “optimal” tuning of algorithm parameters in a practical application will likely not be as great as demonstrated. This should be identified as a limitation of this analysis provided in the paper. Further, the statistical assessment was confusing – was the Bonferroni correction done to account for the omnibus Friedman test since the ANOVA based measures were not possible due to normality issues? Or instead, were they applied to the pairwise comparisons when significance was found. Further, how pairwise comparisons were performed was not states (Wilcoxon signed rank test?). Additional post-hoc corrections should also be applied to pairwise comparisons. Note that there is a non-parametric multi-factor ANOVA option (the aligned rank test) which may assist in performing some of this analysis in conjunction with one another. Specifically, the algorithm and the speed could be analyzed together. Since 10 algorithms were evaluated, I was concerned that the study was under-powered to find changes across them. Finally, was there any interest in investigating KF vs. CF algorithms as categories? I recommend accept with moderate revision.

Comments pertaining to specific sections of the text follow.

  1. The title is quite long and does not mention that the work pertains specifically to human motion capture. Would suggest something like "Comparative analysis of ten inertial and magnetic sensor fusion algorithms for orientation estimation in human motion capture". If a subtitle is desired, a more idiomatic expression would be “one size does not fit all”.
  2. Line 45: “the difference between the orientation of the LCS… and its actual orientation” – It is explained later that this is measured using the minimum rotation angle between the two orientations, but should be stated here too.
  3. Lines 48 to 51: The format of these two definitions is not consistent. One has just the symbol to the left of the colon, and the other has a short definition of the symbol together with the symbol.
  4. Line 52: Not a mathematically accurate use of the word “subspace”, since the space described is not technically a vector space (does not satisfy closure). Would suggest using “range” or “set”. Also, define “relevant”.
  5. Line 129: “identical experimental conditions (different amplitudes of motion at different rotation rates, different commercial MIMUs, …)” – What is this saying? Is it saying amplitudes of motion, rotation rates, IMU model, etc. should be kept the same, or is it saying they are not part of the experimental conditions? This becomes clear later, but the use of “different” to describe something that is “identical” is confusing.
  6. Lines 145 to 147: Expand on the contributions of the paper here to be explicit about why this comparative analysis is needed and what does this paper do that the other comparative studies cited don't do?
  7. Lines 154 to 157: Make clear that the tuning was performed on the same dataset as that used for the evaluation and note that, for this reason, the “best achievable performance” is not what would realistically be achievable in a practical application
  8. Line 160: “selected among the most popular and performing ones” – Provide some more rationale for selection of the SFAs. What is the basis for saying these are the most popular ones? What does each selected algorithm contribute to the analysis compared to the others? Why were only Kalman filters and complementary filters considered (and not e.g. machine learning)?
  9. Lines 160 to 166: A brief description of how each of these work and whether they are a CF or KF would be helpful. For the MATLAB ones, name the MATLAB routine or the package the implementation is found in.
  10. Lines 169 to 170: “trade-off between the search space dimension and the related computational time” – Seems like the latter would be directly correlated with the former. Perhaps “compromise is the word intended?
  11. Lines 170 to 171: “the gyroscope is the main source of information in a sensor fusion framework” – Please clarify/justify this comment. This is not true in general, since it depends on the relative weighting of the accelerometer/magnetometer measurements compared to the gyro measurements.
  12. Lines 171 to 172: “the parameter related to the weight given to it” - Does this refer to the gyroscope noise covariance?
  13. Lines 172 to 173: “All the remaining parameter values are set to default” – For the Kalman filters, these usually correspond to physically meaningful quantities (e.g. covariance of disturbances, sensor noise covariance, etc.) Could these not be derived from the truth data or sensor characteristics? This would provide a closer estimate of these values than the default. Even if this is not done here, it should be mentioned that this is an option for tuning the algorithms.
  14. Table 1: Define “a.u.”
  15. Table 1: Why was only one parameter tuned for some of the algorithms that had more parameters available to tune?
  16. Table 1: There is a wide range of default values for parameters all labeled "inverse gyroscope weight". Does this refer to the same thing in each SFA? A brief explanation of each algorithm as suggested previously may help explain this.
  17. Line 185: “A wooden board was used” – Were any measures taken to ensure the board was not warped?
  18. Lines 209 to 210: “a dynamic recording was executed to manually orient the board by covering the three rotational degrees of freedom” – Provide a more detailed description of the motion that was performed. Was the motion necessarily representative of the type of motion that the SFAs are designed to capture? Also, where was the board held while this action was being performed? Based on Table 1, it seems like some of the algorithms have features that filter out linear acceleration disturbances. If the board was held by one of its ends, then the acceleration disturbances measured by the MIMUs would be greater than expected for a body-worn IMU due to the greater distance from the center of rotation, which results in higher tangential/centripetal acceleration components. Additionally, because the MIMUs are not exactly co-located, each one will measure a different acceleration while the board is rotating.
  19. Lines 295 to 296: See comments above on statistics
  20. Lines 298 to 299: “a single orientation updated iteration” – Is this a single time step within the algorithm, or a single run of the algorithm for the entire dataset under a given experimental condition? Reading on below, it looks like this is for an entire dataset, but also make this clear here.
  21. Line 307: See previous comment regarding the word “subspace”
  22. Line 308: “see section 2.5.1. for the justification of this uncertainty band” – Does this refer to the SP errors? Why not just state this?
  23. Line 310: This notation is technically incorrect since it suggests that the optimal region is always a square. However, upon examining Figure 3 and the figures in Appendix C, this is clearly not the case. A more precise notation would be {(p_{opt_1}, p_{opt_2})} = {(p_1, p_2) | e <= e_opt + 0.5 degrees}.
  24. Lines 328 to 329: Since this is a mobile processor, also state the operating system used and whether any other programs were running concurrently. The clock rate of these processors is highly dependent on the number of parallel tasks running. The base clock rate may be 1.8 GHz, but can increase to just under 5 GHz for a single task.
  25. Table 3: It would help to categorize these by KF and CF as well as number of parameters tuned (1 or 2) to see if there are any interesting trends, even if not statistically significant. Also, avoid orphaning table headers.
  26. Lines 354 to 358: Seems like this should go just after Table 2 or at least prior to the current subsection
  27. Lines 361 to 362: “no statistically significant differences existed among the 10 SFAs under optimal working conditions” – Perhaps the experiment was underpowered. Including results from each individual IMU separately rather than averaging each pair may help increase the power. Also, if the Bonferroni correction was used for pairwise comparisons, with 10C2 = 45 comparisons, Bonferroni may just be too conservative. Could consider using a different correction, e.g. Tukey.
  28. Table 5: The caption suggests that the Friedman test was used for pairwise comparisons, which doesn’t make sense as per my comments above.
  29. Lines 412 to 414: “If the parameter values are optimized for a specific experimental scenario, the same values can lead to large errors when varying the experimental conditions” – While this is likely still true, the magnitude of improvement may not be as dramatic as represented here due to the previously mentioned potential for over-fitting. The way the tuning is done in this study may mask errors attributed to run-to-run variation. This caveat should be noted.
  30. Lines 462 to 465: “Since the accelerometer aids the sensor fusion process by providing the gravity direction information to compensate for the inclination drift, when the gravity recording is corrupted by high values of linear acceleration then the accelerometer contribution becomes detrimental” – This can be verified by examining the heading and inclination components of the absolute orientation errors for each case.
  31. Lines 503 to 504: The fact that the built-in MATLAB routines are slower than the custom implementations is very surprising. Checking for the existence of variables does not impose much computational overhead and does not explain the MATLAB implementation being 32 times slower. Are there any other possible explanations?
  32. Lines 508 to 509: Would this be a more plausible explanation for the difference in computation time between MCF and VAC?

Additionally, the following corrections are suggested.

  1. Line 67: “respect a” -> “respect to a”
  2. Line 75: ‘During static” ends awkwardly
  3. Line 79: “respectively” is unnecessary
  4. Line 101: “looking the” -> “looking at the”
  5. Lines 116 to 117: “and” is unnecessary
  6. Line 140: Line break here is unnecessary
  7. Table 1 caption: “#params column…” -> “The #params column…”, “p_1 and p_2 report…” -> “The p_1 and p_2 columns report…”
  8. Line 208: “which to be removed” -> “which had to be removed”

Reviewer 2 Report

The manuscript is about the comparing of different orientation estimation using inertial sensors. Three types of inertial sensors have been included in this work. 

My comments are as follows:

  1. How do the authors synchronise different inertial sensors? Why use different sampling frequency of different sensors. Clarify what data has been used? Used calibrated data or raw data from the sensors?
  2. More details about how the dynamic recording needs to be included. It would be good to include a figure to further illustrate.

Reviewer 3 Report

Manuscript: “Analysis of the accuracy of ten sensor fusion algorithms for orientation estimation using inertial and magnetic sensing under optimal conditions: a single outfit for every season does not work”.

ID: sensors-1113491

In this paper the authors compared ten algorithms for estimating the orientation of MIMU sensors.

The authors also compared sensors from 3 different vendors and performed motion tests at three different speeds. The sensors were rigidly attached to a wooden board whose motion was measured by means of an optoelectronic system and used for reference.

The main conclusions were: it is not possible to identify the best performing algorithm. Error increases with rotation rate. Different sensors showed different errors. Performance is influenced by external conditions.

The paper fits within the aims of the journal.

Most of the findings of the present studies were already known from previous literature. The main novelty is in the study and tuning of the parameters to be used with the algorithms.

The paper seem to contain many self-citations of the authors.

I think that there are many aspects that should be improved before the paper can be accepted for publication, especially concerning the overall presentation. In the following some detailed comments that may help the authors.

Detailed comments:

In the paper there are several references to external resources. I would suggest to remove these from the paragraphs and place them all together in a section at the end of the manuscript (e.g. supplementary material).

Line 86: there are 21 papers cited in a block (!). Are all those citations necessary?. And again another block at line 94. I would suggest to reduce them and comment/discuss the most significant ones.

Line 166 please check the year of the citation that seems wrong.

Line 167-173: I would suggest to clarify this part. It is not clear how the parameters are chosen or defined.

Experimental protocol:

  • The gyroscope bias was computed through a static trial. But, it is not clear how this was removed from the “dynamic” trials. Was it subtracted? Some studies suggested to high-pass the signal to remove the drift, was this considered?

  • Please clarify how the “RAW” data was obtained from each sensor. I think that this is important because some manufacturers implement some automatic filtering and correction on the data provided by some versions of their software. (This may be a bias for the comparison across sensors).

  • In addition to the previous, please clearly report the characteristics of the sensors (Accuracy, sensitivity, range, snr, etc.). If a model has a lower accuracy, that would trivially explain the larger error observed in the reconstruction of the orientation.

  • The authors chose to express the orientation and the error as quaternions. I would expect the error in orientation expressed in angles (deg.), how was this obtained from the quaternions? I was expecting something like the Euler angles of the variation (3 numbers).

  • Please clarify how the statistical testings were done (what groups were tested, how many samples for each group, sample size (n. Of repetitions), etc.).

  • Clarify the study on the parameters: I could not fully understand how the parameters were changed with respect to the trials and how the comparison was done.

On Page 11 I read that the data distributions were not normal. As this is an experiment in a controlled environment, could the authors increase the number of measurements/repetitions?

The presentation of the results and the discussion are very confusing. Please clarify:

  • The comparison across Algorithms (that seem to be the main aim)

  • The comparison across sensors

  • The comparison across different rotation rates.

A suggestion may be to separate the three problems. Also, maybe table 2 should be presented as histograms

The x-sens software offers the built-in reconstruction of the orientation (with a declared accuracy). I think it should be worth to compare the results of this study to such data.

Please clarify the discussion. Many sentences duplicate the information given in the introduction.

I think that the first 2 paragraphs of the conclusion are not necessary.

Round 2

Reviewer 3 Report

I noticed significant improvements in the manuscript.

Most of the previous comments were addressed.